# Magnetic Resonance Imaging Measurements of the Proximal Palmar Cortex of the Third Metacarpal Bone and the Suspensory Ligament in Non-Lame Endurance Horses before and after Six Months of Training

**DOI:** 10.3390/ani13061106

**Published:** 2023-03-20

**Authors:** Ines Likon, Sue Dyson, Annamaria Nagy

**Affiliations:** 1Equine Department and Clinic, University of Veterinary Medicine Budapest, 2225 Dóramajor Üllő, Hungary; 2The Cottage, Church Road, Market Weston, Diss IP22 2NX, UK

**Keywords:** proximal metacarpal region, exercise, lameness, MRI

## Abstract

**Simple Summary:**

Pain in the region of the origin of the suspensory ligament is a common cause of lameness in endurance horses, but the effect of exercise on the suspensory ligament and its attachment site on the cannon bone has not been described. This study aimed to document magnetic resonance imaging (MRI) changes in the region of the origin of the suspensory ligament induced by six months of endurance training and competition in six non-lame novice and six non-lame experienced endurance horses. Measurements were acquired from low-field MR images. After six months of exercise, there were no changes in the thickness of the cannon bone cortex or the size of the suspensory ligament. In the area where the suspensory ligament starts, the cortex of the cannon bone was thicker in older, more experienced horses than in younger, less experienced horses. This suggests that exercise has a long-term effect. The results of this study can aid veterinarians in the interpretation of MRI studies of lame horses. To establish the long-term effect of endurance exercise on the structures in the region of the suspensory ligament origin, further studies with a larger number of horses and longer follow-ups are needed.

**Abstract:**

Proximal metacarpal injury is common in endurance horses, yet exercise-induced changes in this region have not been described. This study aimed to document objective exercise-induced changes in the proximal palmar cortex of the third metacarpal bone (PcMcIII) and the suspensory ligament (SL). Low-field magnetic resonance (MR) images of both proximal metacarpal regions were obtained from six novice and six experienced horses, before and after six months of endurance training. Measurements were acquired in T1-weighted transverse MR images at four levels and included the thickness of the PcMcIII, the mediolateral width, and the dorsopalmar depth of the entire SL and its lobes. We used *t*-tests or their nonparametric equivalents to compare the measurements from the two examinations and both novice and experienced horses. The medial aspect of PcMcIII was significantly thicker in experienced horses than in novice horses at 2 and 3 cm distal to the carpometacarpal joint. This likely reflects the cumulative effect of long-term exercise and possibly age. The PcMcIII was significantly thicker medially than laterally. There was no significant difference between pre- and post-season measurements. Six months of endurance training were not sufficient to induce changes in the thickness of PcMcIII or the SL that are detectable in low-field MR images.

## 1. Introduction

Lameness is the most common veterinary problem in endurance horses [1,2,3]. Proximal metacarpal pain is one of the most common reasons why endurance horses experience lameness (Figure 1). It is often caused by injuries to the proximal palmar cortex of the third metacarpal bone (PcMcIII) and the proximal aspect of the suspensory ligament (SL) [3]. Abnormalities of these structures are usually diagnosed with radiography and/or ultrasonography. Advanced diagnostic imaging may be warranted when pain causing lameness is localized to the proximal metacarpal region with diagnostic anaesthesia and a definitive diagnosis is not achieved by traditional diagnostic imaging modalities, or the severity of lameness cannot be explained by the observed abnormalities [4,5,6]. Magnetic resonance imaging (MRI) has provided useful additional information for the diagnosis of proximal metacarpal pain, allowing detailed assessment of both the soft tissue and osseous structures [5,6,7]. The most commonly detected MRI abnormalities considered to be contributing to lameness localized to the proximal metacarpal region include osseous abnormalities at the medial palmar aspect of the third metacarpal bone (McIII). For example, thickening, periosteal and endosteal new bone, mineralization in the dorsomedial aspect of McIII or bone resorption, and fractures [6,8,9,10]. Soft tissue changes include loss of structure and/or increased signal intensity within the SL, as well as dorsal margin irregularity of the proximal aspect of the SL and abnormalities in the medial intermetacarpal articulation [6,8,9,10]. Palmar cortical and SL abnormalities are often seen concurrently in horses with lameness localized to the proximal metacarpal region [6,9,10].

Bone structural changes occur in response to mechanical stimulation that is capable of inducing bone deformation [11,12]. This process appears to be highly age- and site-specific. An immature skeleton can adapt to a greater degree than the bones of mature horses, but bone adaptation is a lifelong process [11,12]. Functional adaptation of tendons and ligaments appears to be much less pronounced than that of the bone, especially in mature animals. It has been shown that exercise may advance age-related changes, e.g., disruption of collagenous structure and alterations in extracellular matrix composition [11,12]. This, in turn, leads to the accumulation of microdamage in the tendons and predisposes them to clinical injury. Exercise-induced hypertrophy, reflected by an increased cross-sectional area, is postulated to alter the mechanical properties of these tendons, making them less efficient at storing energy and more susceptible to injury [12,13,14,15]. Even though proximal palmar metacarpal pain associated with SL or PcMcIII pathology is postulated to be one of the main causes of lameness in endurance horses, current scientific literature offers very little information on the response of these structures to exercise. The understanding of the latter is crucial for differentiating exercise-induced adaptive changes from injuries associated with pain and lameness. This study aimed to describe low-field MR measurements of the PcMcIII and the proximal aspect of the SL in non-lame novice and experienced endurance horses, before and after six months of endurance training and competing. We hypothesized that there would be an increase in the thickness of the PcMcIII following six months of training and competing and that the PcMcIII would be thicker and the SL measurements larger in experienced horses than in novice horses.

## 2. Materials and Methods

### 2.1. Data Acquisition

Horses were eligible to participate in the study if they were considered non-lame by their riders and had no history of previous injury in the proximal metacarpal region. Additionally, it was intended to train and compete (if appropriate) the horse in endurance following the first examination. Participation was voluntary and by invitation. Owners gave their written consent for their horses’ participation. The study was approved by the Ethical and Animal Welfare Committee of the University of Veterinary Medicine Budapest PE/EA/00140-4/2022. Twelve endurance horses were selected: six novice horses (that had never competed at >80 km distance) and six experienced horses (that had completed at least two ≥120 km rides). A convenience sample was used, and the first 12 applications that matched the criteria were accepted. All horses were examined clinically twice, once in January or February 2021 (“pre-season”) and again approximately six months later (“post-season”). The examinations were performed by an experienced clinician (AN, Diplomate of the American and European Colleges of Veterinary Sports Medicine and Rehabilitation) and consisted of a clinical examination and gait evaluation in hand and on the lunge on both soft and firm surfaces. Horses were also observed ridden by a skilled technician in an arena. Inclusion criteria stated that only horses that did not show a grade ≥1/8 lameness in any limbs on the initial examination, under any circumstances, were included in the study [16]. Prior to admission to the study, a questionnaire was used to obtain the horses’ history, including questions about any previous injuries, training, and competition (Appendix A). The questionnaires were completed by the rider or trainer prior to, or at the time of, admission, and the data were verified by an in-person interview when horses were admitted for the first examination. Data on the number and distance of competitions and any time off training exceeding two weeks during the study period were collected at the time of the second examination and verified by checking the competition records [17].

Low-field magnetic resonance images of both proximal metacarpal regions were acquired in a 0.27 T open magnet (Hallmarq Veterinary Imaging Ltd., Guildford, UK) using a fetlock coil. The metacarpal region of the imaged limb was positioned as vertically as possible in the center of the magnet. A standard clinical scanning protocol was used. including T1- and T2*- weighted gradient echo (GRE), T2-weighted fast spin echo (FSE), and short tau inversion recovery (STIR) FSE fast sequences in sagittal, frontal, and transverse planes (Table 1). Horses were sedated with a combination of acepromazine (0.04 mg/kg IM, Vetoquinol, Lure Cedex 70204, France), romifidine (0.01 mg/kg IV, Boehringer Ingelheim, Ingelheim, Germany), detomidine (0.01 mg/kg IV, Orion Pharma, Budapest, Hungary), and butorphanol (0.02 mg/kg IV, Bioveta, Ivanovice na Hané, 683 23, Czech Republic). Top-up sedation was administered as required during the scanning procedure.

Measurements were acquired by a single analyst (IL) who received prior training from an experienced assessor (AN, Diplomate of the American and European Colleges of Veterinary Sports Medicine and Rehabilitation, with 15 years of experience in MR image interpretation). Measurements were acquired in T1-weighted GRE transverse MR images at 2, 3, 5, and 7 cm distal to the carpometacarpal joint (CMCJ). The levels were set using a midline sagittal image for cross-referencing (Figure 2). The palmar cortical thickness of McIII was measured at 25%, 50%, and 75% of its mediolateral width (Figure 3A,B). Measurements were obtained perpendicular to the palmar contour of the cortex. The overall palmar cortical thickness was calculated, using Microsoft Excel (Microsoft Corp., 2019; Redmond, Washington, DC, USA), as the mean of the measurements obtained at 25%, 50%, and 75% of the mediolateral width of the PcMcIII, for each level, respectively. Maximum mediolateral width and dorsopalmar depth were measured for each lobe of the SL or for the entire SL, where the lobes could not be distinguished. Dorsopalmar depth was measured as the greatest distance between the dorsal and palmar margins of the ligament, obtained perpendicular to its palmar margin. The mediolateral width was measured as the greatest distance between the medial and lateral margins of the ligament (Figure 3A,B). At 2 cm distal to the CMCJ, the margins of the SL cm were poorly defined in most limbs, and dorsopalmar depth and mediolateral width measurements could not be obtained reliably. Suspensory ligament measurements obtained at this level were therefore excluded from the analysis. All measurements were performed in a dedicated medical image viewer software (Horos Project; https://horosproject.org, accessed on 1 May 2021).

### 2.2. Data Analysis

A repeatability study was performed by IL on a sample of 10 limbs of the first five horses for which images were acquired. Three measurements were obtained for each parameter, and the coefficient of variance was calculated using Microsoft Excel. The repeatability of measurements was confirmed (coefficient of variance ≤ 2%) before acquiring the study measurements.

The data were tested for normality using the Shapiro–Wilk test. A paired samples *t*-test or a Wilcoxon signed-rank test were used to assess the difference between the pre- and post-season measurements of the right and left forelimbs, the medial and lateral lobes of the SL, the mediolateral width and dorsopalmar depth of the SL and its lobes, the medial and lateral aspects of the PcMcIII (25% and 75% of its width), as well as the overall palmar cortical thickness of the PcMcIII from proximal to distal. Independent samples *t*-tests or Mann–Whitney U tests were used to assess the differences between novice and experienced horses. Pre- and post-season data were also pooled and evaluated again for the difference between novice and experienced horses using an independent samples *t*-test or Mann–Whitney U test. Furthermore, the differences between right and left forelimbs, medial and lateral aspects of the PcMcIII (25% and 75% of its width, respectively), overall palmar cortical thickness of the PcMcIII, medial and lateral lobes of the SL, and mediolateral width and dorsopalmar depth of the SL and its lobes were also assessed using a paired samples *t*-test or a Wilcoxon signed-rank test. Statistical analyses were performed in SPSS (IBM Corp., 2019; New Orchard Road Armonk, New York, NY, USA). Statistical significance was set at *p* < 0.05.

## 3. Results

Clinical examination and MRI of the proximal metacarpal region were performed on 12 horses pre-season and 11 horses post-season. One horse was sold during the study period. The mean time between the examinations was 203 days (range: 184–226 days). All horses were Arabians or Arabian crosses, aged 3–17 years (mean 9; median 7.5). Experienced horses were aged between 8 and 17 years (mean 12.7; median 13), while novice horses were between 3 and 7 years old (mean 5.3; median 5.5). Horses weighed 390–483 kg (mean 429.3; median 427) at the first examination. There were eight mares and four geldings. On average, horses completed 1.5 competitions (range: 0–3 competitions) in the 2021 racing season between the two examinations, ranging from 20 to 160 km in distance (mean 65 km; median 70 km). All horses were turned out in a field for at least part of the day. Ten out of 12 horses (83%) were mostly trained on hard (e.g., gravel road), undulating terrain; the remaining two horses were trained predominantly on soft, undulating terrain. In 11/12 horses (92%) training included sections of trotting on asphalt. Most horses (11/12, 92%) were also engaged in some type of unridden work, which included walking in a horse-walker and exercising on the lunge. The majority of horses (9/12, 75%) performed ridden exercise 5 days per week (range 3–5 days). Further details regarding individual horses can be found in Table 2. The mean time off training was 19 days (range: 0–60 days). Most horses had time off following competitions, usually 1/10 of the distance in days (e.g., 16 days after a 160 km competition). Two experienced horses had longer time off: horse 8 due to distal tarsal pain and horse 9 due to a hind foot abscess. Novice horses training towards qualification did not have any time off training.

On the initial examination, no horses showed forelimb lameness or clinical signs of proximal metacarpal pain. Six horses developed lameness between examinations, which was recognized by the trainer or rider. These horses were referred for investigation to the third author; four horses showed forelimb lameness and two horses hindlimb lameness. Only one horse had a small component of proximal metacarpal pain (a greater part of the pain causing lameness was localized to the fetlock region using diagnostic anaesthesia). All horses returned to training and were in full work at the time of the second examination. No horses showed forelimb lameness at the second examination.

In total, 324 images of 46 limbs were analyzed. Signal intensity variations in the McIII and SL will be reported separately. Overall, the images of all limbs were of acceptable quality for clinical diagnosis. The medial aspect of PcMcIII was thicker in experienced horses than in novice horses at 2 cm (*p* < 0.001; MD [mean difference] = 1.1 mm; 95% confidence interval [CI] 0.6, 1.6) and 3 cm (*p* < 0.001; MD = 1.3 mm; 95% CI 0.8, 1.8) distal to the CMCJ (Figure 3A,B). The results for pre- and post-season measurements are presented in Appendix A. Measurements at 25% of the mediolateral width of the PcMcIII were significantly larger than those at 75% of the mediolateral width (Figure 4A–D; Table 3). The overall thickness of the PcMcIII progressively increased from proximal to distal (Figure 4A–D; Table 3). There was no significant average difference between pre- and post-season measurements or between left and right forelimbs.

The mediolateral width of the entire SL was also significantly greater than its dorsopalmar depth at 7 cm distal to the CMCJ (Figure 4A–D; Table 4). The dorsopalmar depth of the lateral lobe was significantly greater than that of the medial lobe at 3 and 5 cm distal to the CMCJ (Figure 4A–D; Table 4). The dorsopalmar depth of the lateral lobe was significantly greater post-season in experienced horses than in novice horses at 3 cm (*p* < 0.025; MD = 1.3 mm; 95% CI 0.2, 2.4) and 5 cm (*p* < 0.005; MD = 1.1 mm; 95% CI 0.4, 1.9) distal to the CMCJ. There was no significant average difference between pre- and post-season measurements or between the left and right forelimbs.

Measurements of the individual SL lobes could be performed reliably at 3 cm distal to the CMCJ; by 5 cm, the lobes were fused in 8/24 (33.3%) limbs. The mediolateral width of both medial and lateral lobes was significantly greater than their dorsopalmar depth (Figure 4A–D; Table 4).

## 4. Discussion

This is the first study to describe MRI measurements of the PcMcIII and the SL in the proximal metacarpal region of non-lame endurance horses in full training and competition. It provides valuable reference data to compare with lame horses. To our knowledge, this is also the first study to perform sequential MRI examinations in performance horses to document MRI changes as a response to exercise. In agreement with our hypothesis, the medial aspect of the PcMcIII was significantly thicker in experienced horses than in novice horses. This shows that long-term exercise can induce the thickening of the proximal aspect of the PcMcIII. This process appears to be site–specific because the difference between novice and experienced horses in the cortical bone thickness of the PcMcIII was only observed at 2 and 3 cm distal to the CMCJ and only in the medial aspect. Site–specific differences in subchondral bone thickness were previously documented in the proximal aspect of the third metatarsal bone [18]. Subchondral bone thickness was significantly greater laterally than medially in both elite competition horses undergoing high-intensity exercise and general–purpose horses undergoing low-intensity exercise.

### 4.1. Thickness of the Proximal Palmar Cortex of the Third Metacarpal Bone

The PcMcIII was significantly thicker on the medial aspect than on the lateral aspect, which is thought to reflect greater loading of the medial aspect of the limb. This is in agreement with the results of a previous study [19] and is also supported by other diagnostic imaging observations. In dorsopalmar radiographs, the opacity of the proximomedial aspect of the McIII is often slightly greater medially than laterally [20]. On scintigraphic images of normal horses, there is more radiopharmaceutical uptake in the medial aspect of the McIII than in the lateral aspect, reflecting greater bone turnover [21]. Furthermore, most stress-related injuries of the PcMcIII occur at or start from the medial aspect of the bone [8,9,10]. In the current study, significant differences between novice and experienced horses were only seen in the medial aspect of the PcMcIII, providing further evidence that the medial aspect of the palmar cortex is affected the most by exercise. The mean difference between the medial and lateral aspects of the PcMcIII was progressively smaller from proximal to distal, suggesting that any effect of uneven mediolateral loading is more pronounced proximally in the proximal half of the metacarpal region.

The overall thickness of the PcMcIII progressively increased from proximal to distal, which is in agreement with previous observations [19]. Although age is also postulated to affect the adaptability of cortical bone to exercise [11,12,22], a robust statistical assessment of the effect of age on PcMcIII thickness was not possible in this study because of the age distribution (all experienced horses were older than 8 years, while all novice horses were younger than 8 years) and the small sample size. There was no significant difference in the thickness of PcMcIII between the left and right forelimbs, which suggests that in non-lame horses there is good left–right symmetry and the contralateral limb can be used for comparison when establishing the clinical significance of the thickness of the PcMcIII. Conformation might influence the loading of the limb and therefore cortical thickness [3,23]. In the current study, no major conformational abnormalities were identified.

Contrary to our hypotheses, six months of endurance training and competition did not induce thickening of the PcMcIII, which may suggest that the duration and/or intensity of exercise was insufficient to induce changes in this time frame. Alternatively, any induced changes could not be detected using low-field MRI in standing horses.

### 4.2. Size of the Suspensory Ligament

The mediolateral width of the SL lobes was significantly greater than their dorsopalmar depth. This has been previously described in the medial lobe [19], but not in the lateral lobe. The dorsopalmar depth of the lateral lobe was significantly greater than that of the medial lobe, which is in agreement with previous observations [19,24]. The dorsopalmar depth of the lateral lobe of the SL was also significantly larger in experienced horses than in novice horses in the post-season measurements at 3 cm and 5 cm distal to the CMCJ. The significance of these findings is unknown. A study on a larger number of horses is needed to confirm whether age- and/or exercise-related hypertrophy in the SL occurs.

### 4.3. Development of Lameness during the Study Period

During the study period, 6/12 horses (50%) developed lameness, which is in accordance with the findings of previous studies [2,3]. In England and Wales, 53% of endurance horses experienced at least one episode of lameness in 12 months [2]. Furthermore, 63% of Italian endurance horses developed lameness during an observation period ranging from 3 to 140 months [3]. Current literature suggests that proximal metacarpal region pain is one of the most common causes of forelimb lameness in endurance horses. However, in the current study, only one horse had a contribution to lameness from proximal metacarpal region pai. Nevertheless, the sample size was insufficient to determine the prevalence of lameness causes, which was also beyond the scope of this study.

### 4.4. Limitations of the Study

The relatively small sample size potentially resulted in a study with low statistical power. Sample size or power calculations could not be reliably performed for this study because no estimated effect of training was available. Sample sizes in studies using live, client-owned animals are limited by financial constraints. Welfare and ethical considerations also support using the smallest possible sample size that can be used to detect meaningful results. Other technical limitations may have decreased the ability of this study to detect more subtle changes than those described. The use of a low-field MRI system in standing horses inherently results in a lower resolution of the images and an increased occurrence of movement artefacts compared with high-field MR images acquired under general anaesthesia. It is also not always possible to position the limb vertically in the magnetic field, which might also have adversely affected the accuracy of measurements. Despite these limitations, the huge advantage of a low-field MR system must not be overlooked; it allows MRI examination of standing horses, which makes the examination accessible for horses in training and competition. Moreover, the resolution of the images, calculated from the field of view and matrix size, and the repeatability of measurements, together with statistically significant differences in measurements between groups, confirm the validity of the results. Horses belonged to three different trainers and had individually tailored training programs, which may have also influenced the results. To discover more subtle changes caused by exercise over a short period of time, it would be beneficial to do additional research on a larger number of horses and employ a more sensitive diagnostic imaging technique. Further investigation of exercise-induced changes over a longer observation period is also warranted.

## 5. Conclusions

In conclusion, the medial proximal aspect of the PcMcIII was thicker in experienced horses than in novice horses, which reflects the effect of cumulative long-term exercise and possibly age. Endurance training and competition for six to seven months were insufficient to induce measurable changes in the PcMcIII or the SL that are detectable with low-field MRI. The results provide reference ranges in non-lame endurance horses and should help in the interpretation of MR images in lame horses with pain localized to the proximal palmar metacarpal region.

## Figures and Tables

**Figure 1 animals-13-01106-f001:**
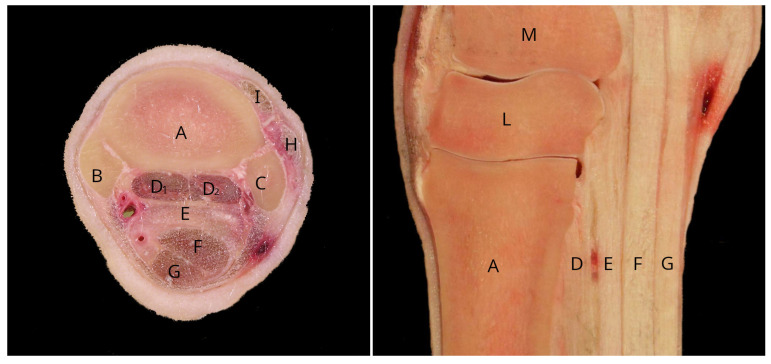
Cross-sectional (**left**) and sagittal (**right**) anatomical specimens of the proximal metacarpal region. (A) third metacarpal bone; (B) second metacarpal bone; (C) fourth metacarpal bone; (D1) medial lobe of the suspensory ligament; (D2) lateral lobe of the suspensory ligament; (E) accessory ligament of the deep digital flexor tendon; (F) deep digital flexor tendon; (G) superficial digital flexor tendon; (H) lateral digital extensor tendon; (I) common digital extensor tendon; (L) third carpal bone; (M) intermediate carpal bone.

**Figure 2 animals-13-01106-f002:**
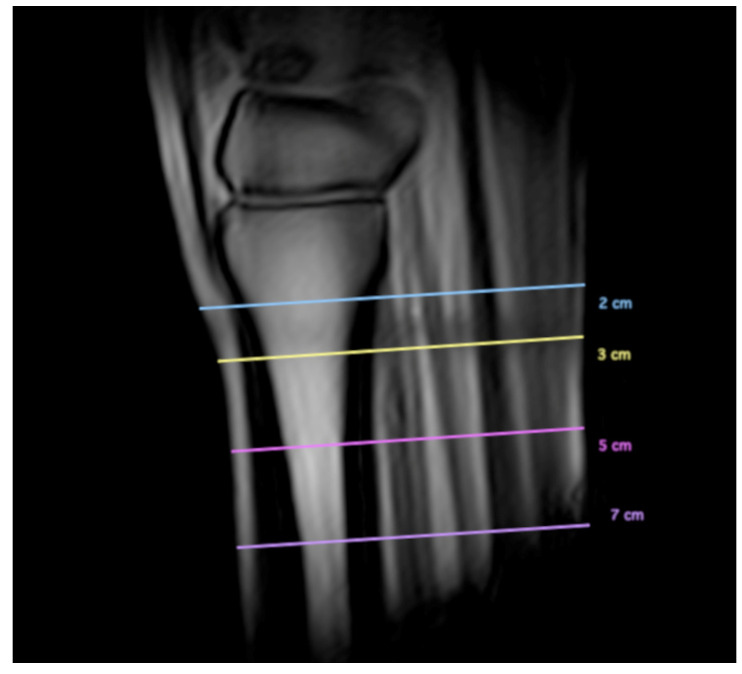
Midline sagittal image from the pilot scan, depicting how the levels were set at 2, 3, 5, and 7 cm distal to the carpometacarpal joint, perpendicular to the suspensory ligament.

**Figure 3 animals-13-01106-f003:**
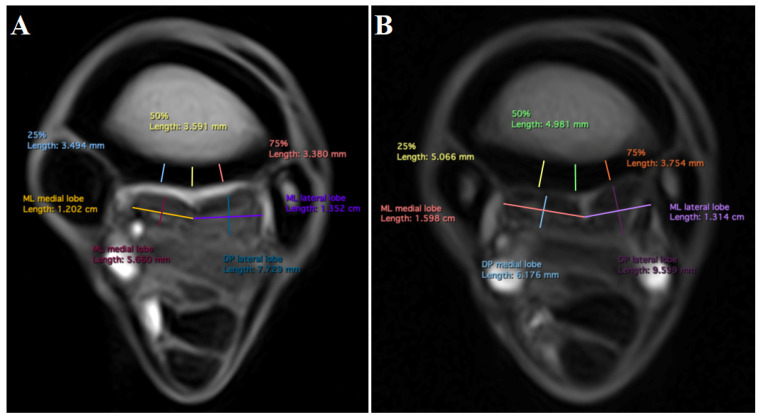
Transverse low-field T1-weighted gradient echo magnetic resonance images of a novice endurance horse (**A**) and an experienced endurance horse (**B**), obtained 3 cm distal to the carpometacarpal joint. Medial is to the left and dorsal is to the top. (ML) mediolateral width, (DP) dorsopalmar depth, and (25, 50, 75%) palmar cortical thickness from medial to lateral.

**Figure 4 animals-13-01106-f004:**
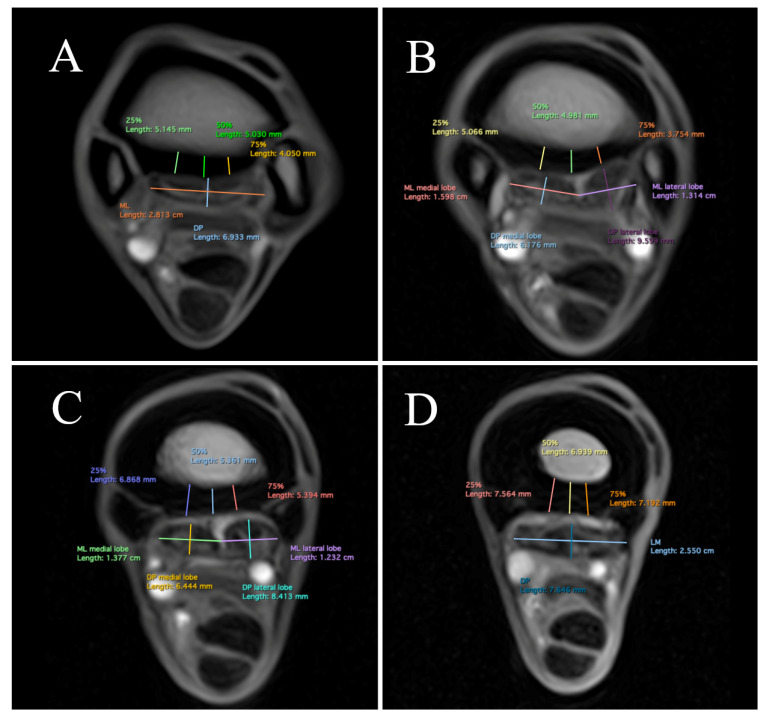
Transverse low-field T1-weighted gradient echo magnetic resonance images of an experienced endurance horse were obtained at 2 cm (**A**), 3 cm (**B**), 5 cm (**C**), and 7 cm (**D**) distal to the carpometacarpal joint. Medial is to the left and dorsal is to the top. (ML) mediolateral width; (DP) dorsopalmar depth; and (25, 50, 75%) palmar cortical thickness from medial to lateral.

**Table 1 animals-13-01106-t001:** Pulse sequence parameters used in the low-field (0.27 T) magnet to image the proximal metacarpal region of 12 endurance horses. The same parameters were used for sagittal, transverse, and frontal sequences. T1W GRE—T1-weighted gradient echo, T2*W GRE—T2*-weighted gradient echo, T2W FSE—T2-weighted fast spin echo, STIR FSE—short tau inversion recovery fast spin echo; TE—echo time; TR—repetition time; FE—frequency encoding; PE—phase encoding; FOV—field of view.

Pulse Sequence	TE [ms]	TR [ms]	Flip Angle [◦]	Slice Thickness [mm]	Slice Gap [mm]	Matrix Size (FE × PE)	FOV [mm]
Pilot	7	66	45	7	0	150 × 120	220
Pilot of a pilot	7	66	45	7	0	150 × 120	220
STIR TEST	22	2100	70/90/120/135	4	0.8	256 × 144	200
T1W GRE FAST	8	52	50	5	1	170 × 170	170
T2*W GRE FAST	13	68	25	5	1	340 × 160	170
T2W FSE FAST	88	1544	90	5	1	168 × 168	170

**Table 2 animals-13-01106-t002:** Signalment and experience of 12 endurance horses and their competitions and days off training during the study period of approximately six months. Experience level was recorded at the start of the study. Novice—horses that had never competed at a distance longer than 80 km at a national level, experienced—horses that had completed at least two ≥120 km international endurance rides, CEN—councours d’endurance national, national-level competitions, CEI—councours d’endurance international, international-level Fédération Equestre Internationale–approved competitions, ***—140–160 km in one day or 90–100 km per day over two days or 70–80 km per day over at least three days, **—120–139 km in one day or 70–89 km per day over two days. For competitions, the completed distance is provided in parentheses if the horse did not complete the full distance.

Horse	Age (Years)	Breed	Gender	Experience Level	Competitions (Distance Completed)	Time off Training	Days Worked per Week	Unridden Work
1	5	Arabian	mare	Novice (no competition)	2 × 40 km, 1 × 80 km,	16 days	5	yes
2	7	Anglo-Arabian	mare	Novice (40 km CEN)	1 × 40 km, 2 × 80 km	28 days	5	yes
3	9	Arabian	gelding	Experienced (CEI ** 120 km)	1 × 100 km	10 days	5	yes
4	6	Arabian	mare	Novice (80 km CEN)	1 × 80 km, 1 × 100 km	18 days	5	yes
5	8	Arabian	mare	Experienced (CEI ** 120 km)	1 × 120 km (sold in July 2021)	sold	5	yes
6	16	Shagya Arabian	mare	Experienced (CEI *** 160 km)	1 × 120 km 1 × 40 km	21 days	5	yes
7	11	Shagya Arabian	mare	Experienced (CEI *** 160 km)	1 × 160 km	16 days	5	yes
8	15	Shagya Arabian	gelding	Experienced (CEI *** 160 km)	160 km (100 km)	35 days	5	yes
9	17	Shagya Arabian	mare	Experienced (CEI *** 160 km)	1 × 60 km	60 days	5	no
10	3	Shagya Arabian	gelding	Novice (no competition)	none	none	4	yes
11	7	Shagya Arabian	mare	Novice (no competition)	3 × 20 km	none	3	yes
12	4	Arabian	gelding	Novice (no competition)	none	none	4	yes

**Table 3 animals-13-01106-t003:** Results of the pre-season (1st examination) and post-season (2nd examination) pooled data analysis for 12 endurance horses examined at an interval of approximately six months, using the paired samples *t*-test or the Wilcoxon signed-rank test for the comparison of measurements of the medial (25%) and lateral (75%) aspects of the proximal palmar cortex of the third metacarpal bone (PcMcIII) (top) and the overall thickness of the PcMcIII (the mean of the measurements obtained at 25, 50, and 75% of the PcMcIII mediolateral width) (bottom), at 2, 3, 5 and 7 cm distal to the carpometacarpal joint (CMCJ). The differences between measurements acquired at 2 cm and 3 cm, 3 cm and 5 cm and 5 cm and 7 cm, respectively, are each compared. NNDD—not normally distributed data, Proximal—the more proximal of the two levels compared (measured in cm from the CMCJ); distal—the more distal of the two levels compared (measured in cm from the CMCJ); SD—standard deviation; MD—mean difference; CI—confidence intervals. * = statistically significant; *p* < 0.05.

PcMcIII 25–75%							
Level	25%		75%		MD [mm]	*p* Value	95% CI [mm]
	Mean [mm]	SD [mm]	Mean [mm]	SD [mm]			
2 cm distal to CMCJ	3.4	1.0	2.7	0.8	0.8	<0.01 *	0.6, 0.9
3 cm distal to CMCJ	4.3	1.1	3.4	0.8	0.9	<0.01 *	0.6, 0.1
5 cm distal to CMCJ	4.7	1.0	4.4	0.8	0.4	<0.01 *	0.2, 0.6
7 cm distal to CMCJ	NNDD ^a^						
Differences in PcMcIII overall thickness between two different levels among the four measured levels							
level	proximal		distal		MD [mm]	*p* value	95% CI [mm]
	mean [mm]	SD [mm]	mean [mm]	SD [mm]			
2–3 cm distal to CMCJ	3.3	0.9	4.4	1.0	1.1	<0.01 *	0.8, 1.4
3–5 cm distal to CMCJ	4.4	1.0	5.0	0.9	0.6	<0.01 *	0.4, 0.8
5–7 cm distal to CMCJ	5.0	0.9	5.8	1.2	0.7	<0.01 *	0.4, 1.0

^a^ Result of Wilcoxon Signed Rank Test: median 25% = 5.2 mm, median 75% = 5.2 mm, *p* = 0.01 *, SE = 77.02.

**Table 4 animals-13-01106-t004:** Results of the pre-season (1st examination) and post-season (2nd examination) pooled data analysis for 12 endurance horses examined at an interval of approximately six months, using the paired samples *t*-test or the Wilcoxon signed-rank test for the comparison of the mediolateral (ML) width and dorsopalmar depth (DP) of the suspensory ligament (SL) and its lobFes (top) and medial and lateral lobes of the SL, at 2, 3, 5, and 7 cm distal to the carpometacarpal joint (CMCJ); NNDD—not normally distributed data; SD—standard deviation; MD—mean difference; CI—confidence intervals. * = statistically significant, *p* < 0.05.

SL ML Width—DP Depth							
Level	ML Width		DP Depth		MD [mm]	*p* Value	95% CI [mm]
	Mean [mm]	SD [mm]	Mean [mm]	SD [mm]			
3 cm distal to CMCJ							
SL medial lobe	14.3	1.8	7.2	1.3	7.1	<0.01 *	6.5, 7.7
SL lateral lobe	13.0	1.0	9.2	1.4	3.8	<0.01 *	3.44, 4.2
5 cm distal to CMCJ							
SL medial lobe	NNDD ^b^						
SL lateral lobe	11.6	1.3	8.0	1.0	3.6	<0.01 *	3.0, 4.1
7 cm distal to CMCJ							
SL total	NNDD ^c^						
SL medial lobe—lateral lobe							
level	medial lobe		lateral lobe		MD [mm]	*p*-value	95% CI [mm]
	mean [mm]	SD [mm]	mean [mm]	SD [mm]			
3 cm distal to CMCJ							
ML	14.3	1.8	13.0	1.0	1.3	<0.01 *	0.7, 2.0
DP	7.2	1.3	9.2	1.4	−2.0	<0.01 *	−2.5, −1.4
5 cm distal to CMCJ							
ML	12.3	1.5	11.6	1.3	0.7	0.7	−0.1, 1
DP	NNDD ^d^						

^b^ Result of Wilcoxon Signed Rank Test: median ML width = 12.4 mm, median DP depth = 7.4 mm, *p* = 0.000, SE = 55.97; ^c^ Result of Wilcoxon Signed Rank Test: median ML width = 22.4 mm, median DP depth = 7.0 mm, *p* = 0.000, SE = 91.53; ^d^ Result of Wilcoxon Signed Rank Test: median DP depth medial lobe = 7.4 mm, median DP depth lateral lobe = 7.7 mm, *p* = 0.008, SE = 55.97.

## Data Availability

Anonymized raw data are available upon reasonable request.

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
