# Peer review of "Magnetic Resonance Imaging Measurements of the Proximal Palmar Cortex of the Third Metacarpal Bone and the Suspensory Ligament in Non-Lame Endurance Horses before and after Six Months of Training"

_animals, 2023, doi:10.3390/ani13061106_

Round 1

Reviewer 1 Report

The article aims to compare the measurements of the palmar cortex of the proximal third metacarpal bone and the proximal suspensory ligament on low-field magnetic resonance images in novice and experienced endurance horses on two occasions before and after competition season. The data shows a symmetry of measurements between both front limbs and could provide a reference to compare to horses with proximal metacarpal pain. The study produces interesting information and a valuable starting point in documenting MRI findings in the proximal metacarpal region in endurance horses with the impact of time and/or training. Older, experienced horses had a thicker cortex than younger, inexperienced horses suggesting the long-term effect of age and/or level of performance and training. There was no significant difference between the measurements before and after competition season, suggesting that a period of 6-7 months is not enough to induce any changes in the proximal suspensory ligament and the proximal cannon bone measurements detectable on standing low-field MRI.

The description of the two groups of horses regarding their lameness status at the time of the two examinations could be more precise and clearer. 

Given the small sample size, statistical power of the study is low and sample size calculations were not done. This was highlighted sufficiently in the discussion. 

Including the description of signal intensity variations would have added to the value of information gained from this paper. 

Measurements were taken by one author. By adding a second author to the repeatability study, information regarding interobserver agreement could have been made.

Simple Summary

·     Please define groups (“six non-lame novice and six experienced horses”): non-lame = sound? Were there lame horses in the experienced group? 

Introduction

·     It would be worth mentioning diagnostic local anaesthesia as an important tool to diagnose proximal metacarpal pain 

Materials and Methods

·     Were horses included with a grade 1/8 lameness? If so, the statement “non-lame novice and experienced endurance horses” at several points in the manuscript would be incorrect. Please clarify.

·     Please provide the reference for the lameness scale used

Results

·     Were all horses sound at the second examination?

·     “Signal intensity variations are reported separately" - in another manuscript?

·     A comprehensive rider/trainer questionnaire was carried out regarding training practice and health. Unfortunately, most of the information from this was not provided in the manuscript. Was the detailed information regarding training considered irrelevant due to the lack of difference between measurements before and after competition season? What was gained from the questionnaire? 

Discussion

·     The following points would be interesting to discuss: time off training (meaning for the results and data interpretation); reasons for horses not to compete despite the intention to do so (how would this have impacted the training regime?)

Figure 2: the font size of the measurements is too small 

Author Response

Comments and Suggestions for Authors

  1. The description of the two groups of horses regarding their lameness status at the time of the two examinations could be more precise and clearer.

We believe that the description of lameness at the first and second examination is clear (Results, p.6, paragraph 1). We do not feel that details of lameness unrelated to the proximal metacarpal region is relevant to this study, but if the Reviewer and the Editor wish, we are happy to provide further details.

“No horses showed forelimb lameness or clinical signs of proximal metacarpal pain on the initial examination. Six horses developed lameness between examinations, which was recognized by the trainer or rider. These horses were referred for investigation to the third author; four horses showed forelimb lameness and two horses hindlimb lameness. Only one horse had a small component of proximal metacarpal pain (a greater part of the pain causing lameness was localized to the fetlock region using diagnostic anaesthesia). All horses returned to training and were in full work at the time of the second examination. No horses showed forelimb lameness at the second examination.”

  1. Including the description of signal intensity variations would have added to the value of information gained from this paper.

Thank you for this suggestion. We are preparing a separate manuscript on subjective MRI findings in both the distal carpal and the proximal metacarpal regions.

  1. Measurements were taken by one author. By adding a second author to the repeatability study, information regarding interobserver agreement could have been made.

Thank you for this observation. In order to assess interobserver agreement, all measurements would have to have been made by more than one assessor. This was not within the scope of this paper; our primary aim was not to provide absolute values but to assess any potential differences in measurements between novice and experienced horses and between the first and second examinations.

Simple Summary

  1. Please define groups (“six non-lame novice and six experienced horses”): non-lame = sound? Were there lame horses in the experienced group?

Non-lame has been added before ‘experienced’. We prefer the term ‘non-lame’ to ‘sound’ as ‘sound’ can refer to the whole health status of the horse, not just to its gait.

Introduction

  1. It would be worth mentioning diagnostic local anesthesia as an important tool to diagnose proximal metacarpal pain.

The text has been adapted accordingly.

“Advanced diagnostic imaging may be warranted when pain causing lameness is localized to the proximal metacarpal region with diagnostic anaesthesia, and definitive diagnosis is not achieved by traditional diagnostic imaging modalities, or the severity of lameness cannot be explained by the observed abnormalities.”

Materials and Methods

  1. Were horses included with a grade 1/8 lameness? If so, the statement “non-lame novice and experienced endurance horses” at several points in the manuscript would be incorrect. Please clarify.

No horses showed forelimb lameness or clinical signs of proximal metacarpal pain on the initial examination, which is further detailed in the results section. We have clarified the Materials and methods to avoid confusion:

Inclusion criteria stated that only horses that did not show a grade ≥1/8 lameness in any limbs on the initial examination, under any circumstances, were included in the study.

  1. Please provide the reference for the lameness scale used.

Please see reference number 17.

Results

  1. Were all horses sound at the second examination?

Clarification has been added to the text, page 6, paragraph 1.

“No horses showed forelimb lameness at the second examination.”

  1. “Signal intensity variations are reported separately" - in another manuscript?

These data will be published separately; please see our comment above.

  1. A comprehensive rider/trainer questionnaire was carried out regarding training practice and health. Unfortunately, most of the information from this was not provided in the manuscript. Was the detailed information regarding training considered irrelevant due to the lack of difference between measurements before and after competition season? What was gained from the questionnaire?

Yes, the Reviewer is right, we do not feel that it is justified to analyze training data due to the lack of difference between examinations and also due to the small sample size. We included the questionnaire for transparency and although we collected a lot of data, we later realized that the study did not have sufficient power for their analysis. We have included more data on training in the Results.

Discussion

  1. The following points would be interesting to discuss: time off training (meaning for the results and data interpretation); reasons for horses not to compete despite the intention to do so (how would this have impacted the training regime?)

Further details have been added to the Results. There were only two horses that did not compete during the study period (aged 3 and 4), they were due to have their first novice competition later in the season and in the beginning of the following year.

“All horses were turned out in a field for at least part of the day. Ten out of 12 horses (83%) were mostly trained on hard (e.g., gravel road), undulating terrain; the remaining two horses were trained predominantly on soft undulating terrain. In 11/12 horses (92%) training included sections of trot on asphalt. Most horses (11/12, 92%) were also engaged in some type of unridden work, which included walking in a horse-walker and exercise on the lunge. The majority of horses (9/12, 75%) performed ridden exercise 5 days per week (range 3-5 days). Further details regarding individual horses can be found in Table 2. The mean time off training was 19 days (range: 0-60 days). Most horses had time off following competitions, usually 1/10 of the distance in days (e.g., 16 days after a 160 km competition). Two experienced horses had longer time off; Horse 8 due to distal tarsal pain, Horse 9 due to a hind foot abscess. Novice horses training towards qualification did not have any time off training”.

Reviewer 2 Report

This article addresses whether the proximal palmar cortex of the MC3 and proximal aspect of the suspensory ligament increase in thickness after six months of training and competing in both novice and experienced endurance horses. The authors evaluate bone and ligament thickness using MRI. This question is relevant to the topic of lameness in endurance horses, and could help with the interpretation of MRI in lame horses with localized pain in the proximal palmar metacarpal region. Conclusions are consistent with evidence, and address whether changes occurred in the 11 endurance horses evaluated (they did not). The authors did find, however, that the palmar cortex of the third metacarpal is thicker in experienced horses, and attribute this difference to exercise, though they note that age may also be a factor. 

General Comments:

Introduction: 

The introduction provides a thorough background on the type of injuries that occur in endurance horses, and could be strengthened by a figure depicting the anatomy at hand. 

Materials and Methods:

Because the focus of this study is exercise-induced changes, the paper would benefit from the inclusion of additional information on the type of exercise that each horse was subject to (which was part of the questionaire). This information could be included in the Materials and Methods section as a table, or as part of the table already included that summarizes the sample. Further information regarding the diet and living situation of each horse would also be a valuable addition (e.g., were the horses kept in stalls?).

Additionally, this section would benefit from a figure showing clear labels on a complete MC3 and on MC3 cross-sections of where measurements were taken. The figures included in the results section (that show the measurements taken) are a bit difficult to read. 

Results:

All supplementary tables showing summary statistics of results would be helpful to include in the results section. If more room is required, the table summarizing the sample used could be moved to Materials and Methods. 

Discussion:

This article could benefit most from including an evaluation of the effect of exercise on changes to the MC3 and suspensory ligament. While I understand that training methods and competition schedules are out of the authors' control, it seems that they have a relatively complete idea of the type of training (including variables such as surface used for training) and of kilometers travelled during competition, and that these variables could be added to the analysis in a more specific way (rather than "experienced vs novice"). This would provide further evidence that the authors could use to back their conclusion that differences between experienced and novice horses are due to exercise, rather than some other factor such as age. 

Author Response

Introduction

  1. The introduction provides a thorough background on the type of injuries that occur in endurance horses and could be strengthened by a figure depicting the anatomy at hand.

An additional figure depicting the anatomy of the proximal metacarpal region has been added.

Materials and Methods

  1. Because the focus of this study is exercise-induced changes, the paper would benefit from the inclusion of additional information on the type of exercise that each horse was subject to (which was part of the questionaire). This information could be included in the Materials and Methods section as a table, or as part of the table already included that summarizes the sample. Further information regarding the diet and living situation of each horse would also be a valuable addition (e.g., were the horses kept in stalls?).

Additional information about the training regime of individual horses has been added to the text and to Table 2.

  1. Additionally, this section would benefit from a figure showing clear labels on a complete MC3 and on MC3 cross-sections of where measurements were taken. The figures included in the results section (that show the measurements taken) are a bit difficult to read.

An additional figure depicting how the levels were set at 2, 3, 5 and 7 cm distal to the carpometacarpal joint, has been added.

Results

  1. All supplementary tables showing summary statistics of results would be helpful to include in the results section. If more room is required, the table summarizing the sample used could be moved to Materials and Methods.

The relevant supplementary tables have been added to the main text.

Discussion

  1. This article could benefit most from including an evaluation of the effect of exercise on changes to the MC3 and suspensory ligament. While I understand that training methods and competition schedules are out of the authors' control, it seems that they have a relatively complete idea of the type of training (including variables such as surface used for training) and of kilometers travelled during competition, and that these variables could be added to the analysis in a more specific way (rather than "experienced vs novice"). This would provide further evidence that the authors could use to back their conclusion that differences between experienced and novice horses are due to exercise, rather than some other factor such as age.

While we agree that such analysis would be very interesting and useful, we do not feel that sample size is sufficient, and our dataset is robust enough. Distance travelled during competition is a relatively small part of all distance covered, and as we do not have complete information on distance completed in training, such analysis would produce misleading results.

Round 2

Reviewer 2 Report

With this second version of their study entitled “Magnetic resonance imaging measurements of the proximal palmar cortex of the third metacarpal bone and the suspensory ligament in non-lame endurance horses before and after six months of training," the authors have made many of the corrections suggested by the reviewers. In particular, supplementary tables of results from statistical analyses have been moved to the Results section, additional figures have been added to the the Introduction and Methods sections, and additional information regarding the training regime of included horses have been added. These additions have improved the manuscript, which I recommend as acceptable to be published in Animals.